# Optimal controller design for reactor core power stabilization in a pressurized water reactor: Applications of gold rush algorithm

H Abdelfattah[1], M Esmail[1], Said A. kotb[2], Mohamed Metwally Mahmoud[3], Hany S. Hussein[4,5], Daniel Eutyche Mbadjoun Wapet[6]*, Ahmed I. Omar[7], Ahmed M. Ewais[3]

1 Faculty of Technology and Education, Electrical Department, Suez University, Suez, Egypt, 2 Reactors Department, Egyptian Atomic Energy Authority (EAEA), Cairo, Egypt, 3 Faculty of Energy Engineering, Electrical Engineering Department, Aswan University, Aswan, Egypt, 4 Electrical Engineering Department, College of Engineering, King Khalid University, Abha, Saudi Arabia, 5 Aswan Faculty of Engineering, Electrical Engineering Department, Aswan University, Aswan, Egypt, 6 National Advanced School of Engineering, Université de Yaoundé I, Yaoundé, Cameroon, 7 Electrical Power and Machines Engineering Department, The Higher Institute of Engineering at El-Shorouk City, El-Shorouk Academy, Cairo, Egypt

* eutychedan@gmail.com

**Data Availability Statement:** All relevant data are within the manuscript.

## Abstract

Nuclear energy (NE) is seen as a reliable choice for ensuring the security of the world's energy supply, and it has only lately begun to be advocated as a strategy for reducing climate change in order to meet low-carbon energy transition goals. To achieve flexible operation across a wide operating range when it participates in peak regulation in the power systems, the pressurised water reactor (PWR) NE systems must overcome the nonlinearity problem induced by the substantial variation. In light of this viewpoint, the objective of this work is to evaluate the reactor core (main component) of the NE system via different recent optimization techniques. The PWR, which is the most common form, is the reactor under investigation. For controlling the movement of control rods that correspond with reactivity for power regulation the PWR, PID controller is employed. This study presents a dynamic model of the PWR, which includes the reactor core, the upper and lower plenums, and the piping that connects the reactor core to the steam alternator is analyzed and investigated. The PWR dynamic model is controlled by a PID controller optimized by the gold rush optimizer (GRO) built on the integration of the time-weighted square error performance indicator. Additionally, to exhibit the efficacy of the presented GRO, the dragonfly approach, Arithmetic algorithm, and planet optimization algorithm are used to adjust the PID controller parameters. Furthermore, a comparison among the optimized PID gains with the applied algorithms shows great accuracy, efficacy, and effectiveness of the proposed GRO. MATLAB\ Simulink program is used to model and simulate the system components and the applied algorithms. The simulation findings demonstrate that the suggested optimized PID control strategy has superior efficiency and resilience in terms of less overshoot and settling time.

**Funding:** The authors extend their appreciation to the Deanship of Scientific Research at King Khalid University for funding this work through Small Groups Project under grant number (RGP.1/86 /44). This fund has a great role in study design, data collection and analysis.

**Competing interests:** The authors have declared that no competing interests exist.

**Abbreviations:** $k_P$, Proportional gain; $k_I$, Integral gain; $k_D$, Derivative gain; $Pd(t)$, Desired Power; $Pa(t)$, Output Power; A, Area of heat transference between fuel and coolant; C, Precursor concentration; $C_{PC}$, Heat coolant capacity; $C_{PF}$, Capacity heat of fuel; Fr, Power from the fuel component; h, Average total heat transference coefficient; $M_C$, Flow rate of Coolant mass; M, Coolant mass for two fluid nodes; $M_{Cl}$, Water mass of cold leg; $M_F$, Fuel mass for each node; $\rho$, Total reactivity; $\rho_{ex}$, External reactivity; $\tau_{hl}$, The time constant of hot leg; $M_{hl}$, Water mass of hot leg; A, Neutron of generation time; $M_{lp}$, The lower plenum of water mass; $M_{mo}$, Coolant node lump; $M_{up}$, Water mass of the Upper plenum; P, Power in the reactor core in each node; $T_{cl}$, The temperature of the cold leg; $T_{f1-3}$, Temperatures of Fuel in nodes (1–3); $T_{hl}$, Temperature of hot-leg; $T_{lp}$, Temperature of fluid in lower plenum; $T_{m\,1-6}$, Temperatures of moderator in nodes; $T_{up}$, Temperature of Fluid in the Upper plenum; $T_{po}$, Outlet temperature in primary water; $\tau_{cl}$, The time constant of the cold leg; $k_{d1}$, Derivative fuzzy gain; $\alpha_c$, Coolant coefficient of reactivity; $\alpha_f$, Fuel coefficient of reactivity; $\beta_t$, Totally delayed neutron group fraction; $\tau_{up}$, The time constant of the upper plenum; $\tau_{lp}$, The lower plenum time constant; $\tau_C$, Moderator nodes time constant; $\gamma$, The average of six group decay constant.

# 1. Introduction

## a) Motivation and challenges

The current energy and ecological economics study's main focus is the increasing concern about sustainable advances, renewable energy, and supportable environmental practices. The burning of fossil fuels (FFs) to produce power is the main driver of carbon environmental damage, which is what is causing climate change globally [1,2]. Nuclear energy (NE) stands as an inescapable viable alternative for growth sustainability, given the increasing costs of oil and natural gas along with the significant adverse impacts brought on by the utilization of coal. With just a few nuclear fuels, NE power plants can provide a significant amount of power. For example, a coal-fired power plant of 1000 MW capacity burns around 2 million tonnes of coal annually, whereas an NE plant of the same output needs about 190 tonnes of natural uranium as fuel. Utilizing NE would drastically reduce the usage of FFs and lessen the ecological burden caused by coal for the reason that it is clean and produces no greenhouse gases (GGs) [3,4].

Many nations are looking for clean energy sources to meet their increasing energy needs due to the increase in $CO_2$ emissions and the unstable pricing of FFs. The most important worldwide goals for attaining countries' growth potential include lowering $CO_2$ emissions from power generation and additional industrial operations and safeguarding electricity supplies. Utilizing NE power specifically can cut prospective $CO_2$ pollution while ensuring energy security and sustained economic expansion [5,6]. Considering that nearly two-thirds of GGs emissions come from the generation and use of energy, which are essential for achieving the penalty area of sustainable progress. As a consequence, we want to employ NE and globalization aspects to look into the origins of $CO_2$ pollution. Its specific advantages in lowering $CO_2$ pollution have been hotly contested and endorsed, and it is mainly recognized as an environmentally friendly source of energy. Contrarily, it is demonstrated that NE is not actually helping to reduce $CO_2$ pollution and is bound to lose favor because of the dangers of NE explosions and the effects of radioactive materials on the planet [5,7]. NE consumption is therefore regarded as one of the best effective energy supplies and a key energy strategy element for sustainable development as a result of the environmental danger of nonrenewable generators due to its strong social and financial advantages. In conjunction with outside factors like fluctuations in the price of oil, NE generation has steadily risen due to global energy diversification efforts and NE's financial viability. Many nations use NE and reduce their dependency on unpredictable imported FFs to achieve energy and environmental security because it produces affordable power whose cost is not as impacted by volatility in FF prices as coal or gas [8–10].

In light of ongoing environmental and societal developments in these nations, the top 10 NE consumer nations—Canada, China, France, Germany, Russia, South Korea, Sweden, Ukraine, the UK, and the USA—no longer consume as much. These nations have industrialised economies, with 58%, and 60% of the world's energy consumption in 2018, and 2020, respectively. Since they account for over 56 percent of the world's $CO_2$ pollution, these nations have seen a considerable decline in their environmental conditions. The yearly trajectory of $CO_2$ pollution for the top NE consumption nations was presented [11–13].

## b) Literature review and research gap

To demonstrate innovative security measures and the utilization of thorium for the economical NE era, the alternative heavy water reactor (AHWR) is used as an energy-producing device. The AHWR can produce the required power to address the issue of future energy sustainability thanks to the vast thorium supplies that are there [14]. The AHWR is a 300 MW vertical

pressure tube reactor with heavy water moderation and light water cooling. It has a variety of one-of-a-sort technologies, plus a number of passive safety presents and intrinsic safety characteristics. AHWR uses thermal waste and industrialized steam to produce desalinated water. However because AHWR is complex and nonlinear in nature, the settings of the reactor vary in actual time according to the produced power [15]. Reactor power is currently implemented under base-load operating parameters, which is also essential for security concerns. As a consequence, the control scheme constructed should assure the performance indicator, be robust, and be simple to modify. Conventional controls have the advantage of being easy to develop and use, but their performance is typically subpar. The conventional controller PID is evaluated in the presence of a sizable load shift. As a result, in the recent past, a variety of innovative control systems have arisen in an effort to attain excellent NE power plants (NEPPs) performance during operation. The load-following regulation of a nuclear reactor core is crucial. Handling the axial transfer of power as opposed to radial power distribution is the key challenge in researching atomic core power dispersion control [16,17].

The control systems of NEPPs execute the controls of an NE reactor and its power system, equipment, process, and parameters. The security, dependability, and economy of NPPs are clearly affected by the performance of the control scheme. To enhance the performance of the control schemes of NPPs, new control theories, technologies, and approaches have been studied and investigated extensively. The application of these new control approaches can lead to better control system performance, which is essential for ensuring the safety and reliability of NPPs [18]. Through the years, numerous major NE reactors, including AHWR, PHWR, and PWR, were constructed and designed for the production of demand-side electricity. However, until then, NE reactor technology has developed. However, because of these systems' complicated multi-input multi-output coupled architecture and nonlinear behavior, altitudinal oscillation develops in the NE's core. Thus, implementing adaptive control systems is necessary in order to reduce the changes in demand for electricity. Utilizing an appropriate control approach to the demand power variation is the key problem [16,19]. A reliable PID controller for PWR-sort NE reactor power level management was created. In order to prevent power changes, Lyapunov stability evaluation was taken into account when building a disturbance rejection mechanism. The majority of manufacturing processes use PI and PID to control systems owing to their straightforward design and simplicity in installation. However, systems that are real-time may be realistically represented by expanding the controller's degrees of freedom, where the fractional-order (FO) operator can be used in lieu of the I and D control actions' integer-order equivalents. The FOPI, FOPD, and FOPID were used in nonlinear schemes as reliable controllers due to their enhanced degree of freedom [20,21].

It takes a lot of work to recommend different tuning techniques for creating a robust controller for an installation that performs better than its existing controller. Researchers have studied a number of PID gain adjustment methods up until now [22,23]. Additionally, PID gains were adjusted using the genetic method (GM) in order to control the intended plant reaction. The controller values were adjusted utilizing multi-objective GM for the time-domain limitations, like steady-state error (SSE), in order to design an effective control structure for a particular linear scheme. The controller values have been manually modified via the Ziegler Nichols tuned approach to improve the controller performance [24]. A NEPP's load-following (LF) regulation was important. The administration of the axial distribution of energy was the key problem; the control of radial power dispersion of an NEPP had little impact. This control method gave the LF operation's rapid shift in neutron flux a greater scientific foundation [16,25].

In recent years, there has been a major rise in the modeling of robust control approaches as well as the implementations of controllers, which have been implemented into many intricate control systems. In order to construct an adaptive control strategy that can monitor the power

produced more precisely and meet the Lyapunov stability criteria, a research NE reactor's kinetic power model was employed. This formulation of mixed-integer linear programming was used for NEPP analysis and optimization because it offered great flexibility in solving a variety of issues. integrating a model-free active disruptive control method for contrasting the controller's results with the conventional PID with reactive change and an extensive variety of power modifications in NEPP has been proposed as a solution to the issues that have emerged as a result of xenon fluctuations in the NE reactor core and the ranging reactivity in the core [14,26]. The PSO-PID controller offered greater robustness than the GM-PID controller when incorporated into the NPK model of the NE reactor for LF operations of PWR [27].

Hence, recent works have been conducted on the control algorithm of NE reactor power [28]. To address the automatic generation control (AGC) issue in interconnected power systems, a new optimal combined fuzzy PID control strategy was introduced, utilizing the dragonfly algorithm (DA) [29]. The frequency stability of a hybrid two-area multi-sources was supported by employing an enhanced algorithm called eagle strategy arithmetic optimization (ESAO) to choose optimal values for the proposed FOPID controller parameters, leveraging the exploration and exploitation strategies of the algorithm [30]. In a case study of a wind-power plant, an Arithmetic optimization-based MPC was utilized to jointly control voltage and frequency [31]. A innovative hybrid arithmetic-trigonometric optimization approach, employing various trigonometric affairs, was presented for real-time problems with complex and continuously evolving characteristics [32]. To assess the adequacy of finite element simulation against field data, a new stochastic optimizer known as the planet algorithm (PA) was employed. Furthermore, the PA was utilized to determine the optimal parameters of extreme learning machines in order to enhance their ability in diabetes diagnostics [33].

### c) Contributions

In this article, an efficient approach for optimizing PID controller parameters is described. The optimization algorithm used is the gold rush optimizer (GRO). GRO, which emulates the gold prospecting techniques employed during the GR Era, encompasses three fundamental aspects of gold prospecting: migration, collaboration, and panning [34]. These concepts have found application in various domains. For instance, in this work, the GRO approach is implemented to enhance weight and displacement performances, offering robustness and efficacy in the face of uncertain conditions [35]. Furthermore, the researchers in [35] introduce a function that combines mode shapes and natural frequencies, aiming to minimize structural damage. They employ the GRO optimization technique in conjunction with this function to rapidly identify damaged truss structures reliably and consistently. This inspired us to propose the utilization of GRO for estimating the optimum parameters of a PID controller.

The primary goal of this research is to propose optimum PID control settings for regulating reactor power levels during constant and variable input power conditions for a nuclear power reactor based on a WR scheme. This paper introduces an optimal value of PID control gains based on the GRO for the NE reactor model. A comparison between the suggested PID tuning parameters based on the GRO and the PID gains based on PA, AOA, and DA optimization methods. The comparison highlights the superiority of the presented GRO-PID controller.

### d) Paper organization

This paper is organized as follows: Section 2 introduces an explanation of the NE reactor simulation. Section 3 explains the concept of DA, AOA, and PA techniques, and the mathematical model and concept of the GRO are stated. In section 4, the discussions of the obtained results is presented. Ultimately, the conclusion of this work is introduced in Section 5.

## 2. Nuclear reactor design

This study incorporates a dynamic model of a NE reactor, which encompasses the reactor core, upper and lower plenums, and the piping connecting the steam alternator and reactor core. To enhance the accuracy of the reactor design, lumped parameter paths are employed based on fundamental principles. The heat transfer design employed in this procedure aligns with Mann's design for heat exchangers. The fuel elements are separated into 3 nodes, while the coolant is divided into 6 nodes. The design is regularized depending on the representation shown in Fig 1 [36]. Additionally, the model includes the upper plenum, lower plenum, hot

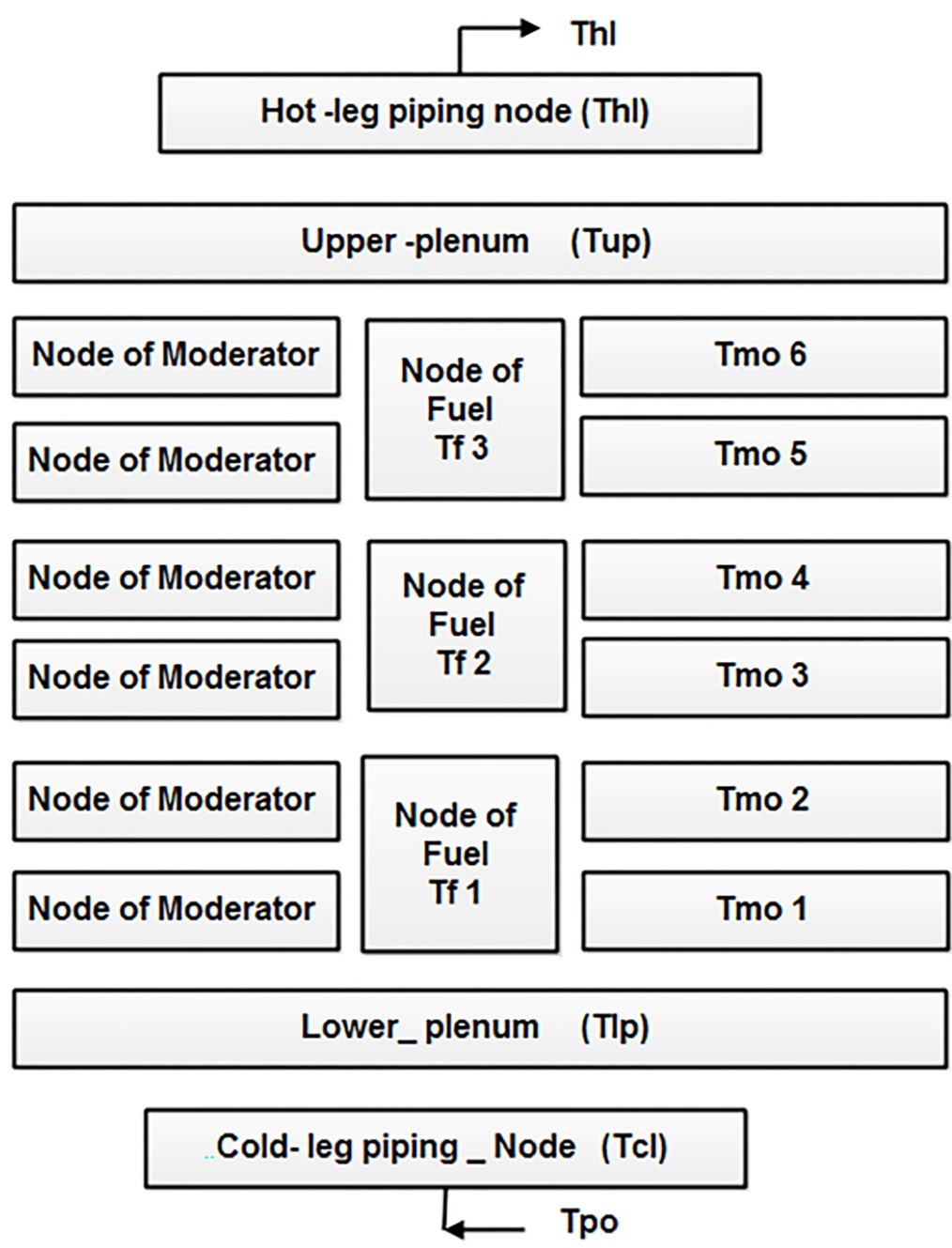

**Fig 1. The diagram of the reactor's moderator design node, containing the primary piping.**

leg, and cold leg as nodes within the reactor coolant scheme. To estimate the average fission power within the reactor core, six delayed neutron designs of point kinetics through average precursor groups are utilized, considering a stable and constant power assumption along the axial path of the reactor core [36]. The algebraic representation of the NE reactor core comprises a group of fifteen differential equations as defined in [36].

Equations of the reactor point kinetics can be written as:

$$\frac{d(P/P_o)}{dt} = \frac{\rho - \beta_t}{\Lambda}\frac{P}{P_o} + \lambda c \tag{1}$$

$$\frac{dC}{dt} = \frac{\beta}{\Lambda}\frac{p}{p_o} - \lambda c \tag{2}$$

The reactor core heat transfer equations

First node:

$$\frac{dT_{f1}}{dt} = \frac{F_r P_o}{(MC_p)_F}\frac{P}{P_o} + \frac{hA}{(MC_p)_F}\left(T_{mo1} - T_{f1}\right) \tag{3}$$

$$\frac{dT_{mo1}}{dt} = \frac{(1 - F_r)P_o}{(MC_p)_C}\frac{P}{P_o} + \frac{hA}{(MC_p)_C}\left(T_{f1} - T_{mo1}\right) + \frac{(T_{lp} - T_{mo1})}{\tau_C} \tag{4}$$

$$\frac{dT_{mo2}}{dt} = \frac{(1 - F_r)P_o}{(MC_P)_C}\frac{P}{P_o} + \frac{hA}{(MC_P)_C}\left(T_{f1} - T_{mo1}\right) + \frac{(T_{mo1} - T_{mo2})}{\tau_C} \tag{5}$$

Second node:

$$\frac{dT_{f2}}{dt} = \frac{F_r P_o}{(MC_p)_F}\frac{P}{P_o} + \frac{hA}{(MC_P)_F}\left(T_{mo3} - T_{f2}\right) \tag{6}$$

$$\frac{dT_{mo3}}{dt} = \frac{(1 - F_r)P_o}{(MC_p)_C}\frac{P}{P_o} + \frac{hA}{(MC_p)_C}\left(T_{f2} - T_{mo3}\right) + \frac{(T_{mo2} - T_{mo3})}{\tau_C} \tag{7}$$

$$\frac{dT_{mo4}}{dt} = \frac{(1 - F_r)P_o}{(MC_p)_C}\frac{P}{P_o} + \frac{hA}{(MC_p)_C}\left(T_{f2} - T_{mo3}\right) + \frac{(T_{mo3} - T_{mo4})}{\tau_C} \tag{8}$$

Third node:

$$\frac{dT_{f3}}{dt} = \frac{F_r P_o}{(MC_p)_F}\frac{P}{P_O} + \frac{hA}{(MC_p)_F}\left(T_{mo5} - T_{f3}\right) \tag{9}$$

$$\frac{dT_{mo5}}{dt} = \frac{(1 - F_r)P_o}{(MC_p)_C}\frac{P}{P_o} + \frac{hA}{(MC_p)_C}\left(T_{f3} - T_{mo5}\right) + \frac{(T_{mo4} - T_{mo5})}{\tau c} \tag{10}$$

$$\frac{dT_{mo6}}{dt} = \frac{(1 - F_r)P_o}{(MC_p)_C}\frac{P}{P_o} + \frac{hA}{(MC_p)_C}\left(T_{f3} - T_{mo5}\right) + \frac{(T_{mo5} - T_{mo6})}{\tau c} \tag{11}$$

Cold leg temperature

$$\frac{dT_{cl}}{dt} = \frac{(T_{po} - T_{cl})}{\tau_{cl}} \tag{12}$$

The lower and upper plenum temperatures can be determined as:

$$\frac{dT_{lp}}{dt} = \frac{(T_{cl} - T_{lp})}{\tau_{lp}} \tag{13}$$

$$\frac{dT_{up}}{dt} = \frac{(T_{mo6} - T_{up})}{\tau_{up}} \tag{14}$$

Hot leg temperature

$$\frac{dT_{hl}}{dt} = \frac{(T_{up} - T_{hl})}{\tau_{hl}} \tag{15}$$

Constitutive equations

$$\rho = \rho_{ex} + \frac{\alpha_f}{3}\left[(T_{f1} + \ldots + T_{f3}) - (T_{f1_o} + \ldots + T_{f3_0})\right]$$
$$+ \frac{\alpha_c}{6}\left[(T_{mo1} + \ldots + T_{mo6}) - (T_{mo1_0} + \ldots + T_{mo6_0})\right] \tag{16}$$

$$\lambda = \frac{\beta_t}{\sum_{i=1}^{6}\frac{\beta_i}{\lambda_i}} \tag{17}$$

$$\tau_C = \frac{M_C}{2\dot{M}} \tag{18}$$

$$\tau_{cl} = \frac{M_{cl}}{\dot{M}} \tag{19}$$

$$\tau_{lp} = \frac{M_{lp}}{\dot{M}} \tag{20}$$

$$\tau_{up} = \frac{M_{up}}{\dot{M}} \tag{21}$$

$$\tau_{hl} = \frac{M_{hl}}{\dot{M}} \tag{22}$$

## 3. Nuclear reactor investigated control algorithms

The three parameters of the PID controller ($K_P$, $K_I$, $K_D$) are optimized in this work using GRO to boost the NE reactor's performance during load-following operation. In order to gauge how well the suggested optimization method performs, a comparison was provided between DA, AOA, PA, and GRO algorithms.

As shown in Eq (23), the objective function (OF) is a set of time-domain integral performance indices. The main purpose is to estimate the minimum values of an index function in order to determine the best PID control strategy settings. The integral of time-weighted

absolute error (ITAE) motivates the selection of an objective function that seeks to minimize the temporal response characteristics.

$$J_1 = ITAE = \int_0^{Ts} t|e(t)|dt \tag{23}$$

where Ts is the simulation time.

### 3.1 Dragonfly algorithm (DA)

A novel optimization technique based on swarm intelligence is introduced, identified as the DA. The inspiration behind the DA, proposed in 2015 [23], arises from the observation of static and dynamic swarming behaviors. These behaviors closely resemble the two key phases of optimization in meta-heuristics: exploration and exploitation. In the exploration phase, dragonflies appear in sub-swarms and traverse various regions, imitating the static swarm behavior. Conversely, during the exploitation phase, dragonflies wing in larger swarms with a unified path, which promotes efficient exploitation of the search space. To simulate the swarming behavior of dragonflies, the DA utilizes 3 fundamental principles of insect swarming proposed in [24], along with two additional concepts: departure, orientation, structure, magnetism to food sources, and distraction from enemies. These five concepts enable the simulation of dragonfly behavior in both dynamic and static swarms. The DA approach is exploited within the framework of particle swarm optimization (PSO), employing two primary vectors: the step vector and the position vector. Those vectors stock the change directions/ speed and position of the dragonflies, respectively.

### 3.2 Arithmetic optimization algorithm (AOA)

AOA leverages the division behavior of key arithmetic operators in math, namely multiplication (M), division (D), subtraction (S), and addition (A). AOA is formulated and carried out as a mathematical model, enabling optimization procedures across a broad variety of search spaces. To demonstrate its versatility, The performance of AOA is assessed using twenty-nine benchmark functions and a variety of real-world engineering design issues. The optimization process in AOA comprises two essential components: exploration and exploitation [37]. AOA boasts a straightforward and uncomplicated implementation, aligning with its mathematical representation, and facilitating its adaptation to address novel optimization problems. It requires minimal parameter adjustments, primarily focusing on population size and stopping criterion, which are standard parameters across optimization approaches. The use of random and adaptive factors improves the divergence as well as convergence of AOA search results [37].

### 3.3 Planet algorithm (PA)

Introducing a meta-heuristic approach called the Planet Approach (PA), informed by Newton's gravitational law. The PA emulates the movement of planets within the solar scheme, with the Sun serving as the central element of the search space. To enhance accuracy and expand the search space concurrently, the PA incorporates two primary phases: local search and global search. To improve the precision of the algorithm, a Gauss distribution function is utilized as an approach [38]. The algorithm commences by setting initial values for the parameters and generating a set of agents, denoted as X, as displayed in Fig 2 [38].

### 3.4 Gold rush optimizer (GRO)

In this section, we present the GRO approach, a population-based metaheuristic approach that emulates the gold prospecting techniques employed during the GR era. The GRO algorithm incorporates three fundamental concepts of gold prospecting: relocation, cooperation, and criticizing. We will outline the mathematical designs for gold prospecting: relocation, cooperation, and criticizing, followed by an explanation of the GRO metaheuristic algorithm [34].

**3.4.1. Gold prospectors modeling.** The GRO approach replicates the significant occurrences of the gold rush period. The positions of gold prospectors are stored in a matrix referred to as $M_{GP}$, represented by Eq (24). In this equation, $x_{ij}$ represents the position of prospector $i$ along the jth dimensions. The variables $d$ and $n$ denote the dimension size and the number of gold prospectors, respectively.

$$M_{GP} = \begin{bmatrix} x_{11} & x_{12} & \dots & x_{1d} \\ x_{21} & x_{22} & \dots & x_{2d} \\ \vdots & \vdots \ddots & \ddots & \vdots \\ x_{n1} & x_{n2} & \dots & x_{nd} \end{bmatrix} \tag{24}$$

To assess the performance of gold prospectors during the optimization process, an objective function is required. The assessment values of gold prospectors are saved in an assessment matrix denoted as $M_F$, as described by Eq (25). In the equation, $x_{ij}$ represents the position of prospector $i$ along the jth dimensions, and $f$ denotes the assessment function.

$$M_F = \begin{bmatrix} f(x_{11} & x_{12} & \dots & x_{1d}) \\ f(x_{21} & x_{22} & \dots & x_{2d}) \\ \vdots & \vdots \ddots & \ddots & \vdots \\ f(x_{n1} & x_{n2} & \dots & x_{nd}) \end{bmatrix} \tag{25}$$

**3.4.2. Migration of prospectors.** Gold prospectors go to it in search of gold. During the process of the metaheuristic algorithm, the optimum point in search space is the precise spot of the richest gold mine. Because the specific site is unknown, the best gold prospector's position is utilized as a starting point for the precise spot of the best gold mine. Eqs (26) and (27) are used to simulate the journey of a gold prospector to a gold mine.

$$\rightarrow D_1 = \rightarrow C_1 . \rightarrow X_i^*(t) - \rightarrow X_i(t) \tag{26}$$

$$\rightarrow Xnew_i(t+1) = \rightarrow X_i(t) + \rightarrow A_1 . \rightarrow D_1 \tag{27}$$

where $\rightarrow X^*$, $\rightarrow X_i$ and $t$ represent the best gold mine site, gold prospector location i, and present iteration t, respectively. $\rightarrow Xnew_i$ represents the additional position of the gold prospector $i$, and $\rightarrow A_1$, $\rightarrow C_1$ represent vector coefficients calculated by

$$\rightarrow A_1 = 1 + l_1(\rightarrow r_1 - \frac{1}{2}) \tag{28}$$

$$\rightarrow C_1 = 2 \rightarrow r_2 \tag{29}$$

where $\rightarrow r_1$ and $\rightarrow r_2$ represent random vectors having values between[0, 1]. $l_1$ is the convergence component termed by Eq (30); If $e$ is equivalent to one, it reduces linearly from 2 to

$\frac{1}{Max_{iter}}$, but it drops nonlinearly for values greater than one.

$$l_e = \left(\frac{max_{iter} - iter}{max_{iter} - 1}\right)^e \left(2 - \frac{1}{max_{iter}}\right) + \frac{1}{max_{iter}} \tag{30}$$

**3.4.3. Gold mining (gold panning).** Each gold prospector digs gold locations in order to discover additional gold. The position of each gold prospector is treated as an estimated place of a gold mine for algebraic modeling [34]. The mathematical relationships important to gold mining are seen as:

$$\rightarrow D_2 = \rightarrow X_i(t) - \rightarrow X_r(t) \tag{31}$$

$$\rightarrow Xnew_i(t+1) = \rightarrow X_r(t) + \rightarrow A_2. \rightarrow D_2 \tag{32}$$

where $\rightarrow X_r$, $\rightarrow X_i$, $t$ and $\rightarrow Xnew_i$ denote the place of a randomly chosen gold prospector r, the position of the gold prospector $i$, present iteration $t$, and the new position of the gold prospector i, respectively. $\rightarrow A_2$ is the vector constant computed by Eq (33). In this equation, the parameter $l_2$ is utilized instead of the parameter $l_1$ to raise the exploitation capability of the mining technique.

$$\rightarrow A_2 = 2l_2 \rightarrow r_1 - l_2 \tag{33}$$

**3.4.4. Collaboration between prospectors.** The algebraic modeling of Eqs (34) and (35) is utilized to explain the cooperation between gold prospectors where $g_1$ and $g_2$ represent two randomly chosen gold prospectors. In this instance, three-person cooperation is realized among prospectors $i, g_1, g_2$, and $\rightarrow D_3$ is the collaboration vector.

$$\rightarrow D_3 = \rightarrow X_{g_2}(t) - \rightarrow X_{g_1}(t) \tag{34}$$

$$\rightarrow Xnew_i(t+1) = \rightarrow X_i(t) + \rightarrow r_1. \rightarrow D_3 \tag{35}$$

If the value of the objective function improves, the gold prospector upgrades its location; otherwise, it remnants at the prior place, which is represented by Eq (36) in elimination issues. Fig 3 depicts the GRO method, which is both intuitive and thorough.

$$\rightarrow X_i(t+1) = \rightarrow Xnew_i(t+1) \quad \text{if } f(\rightarrow Xnew_i(t+1)) < f(\rightarrow X_i(t)) \tag{36}$$

## 4. Results and discussions

Two test scenarios are provided to evaluate the suggested controller's efficacy. The first scenario involves constant reference power, whereas the second involves fluctuating reference power. A comparison through four various optimization methods is introduced to judge the presented GRO-tuned PID controller. These four optimizations-PID controllers are DA, AOA, PA, and GRO. In this study, the reactor core of the PWR scheme is utilized simultaneously through the controllers shown utilizing MATLAB/SIMULINK. The data utilized for the model are provided in [36]. The studied system values are stated in Table 1.

In terms of dynamic performance considerations, the behavior of the system with the provided optimizing and tuning PID controller would be assessed by the profile of core power, fuel and moderator temperatures, and hot leg temperatures. This includes steady-state inaccuracy, maximum overshoot, and settling time. The selection of PID parameter ranges has been

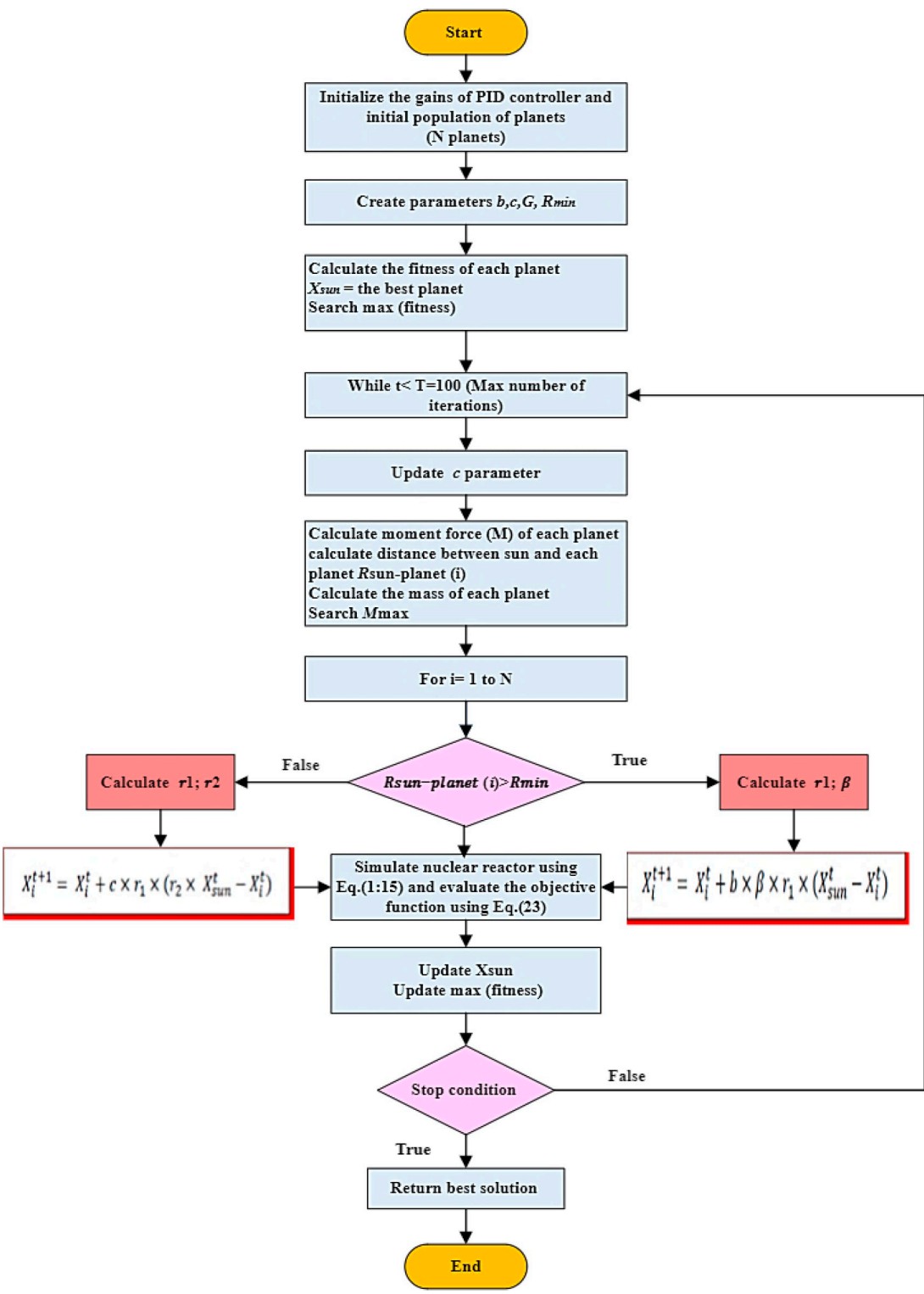

**Fig 2. Flowchart of the PA approach.**

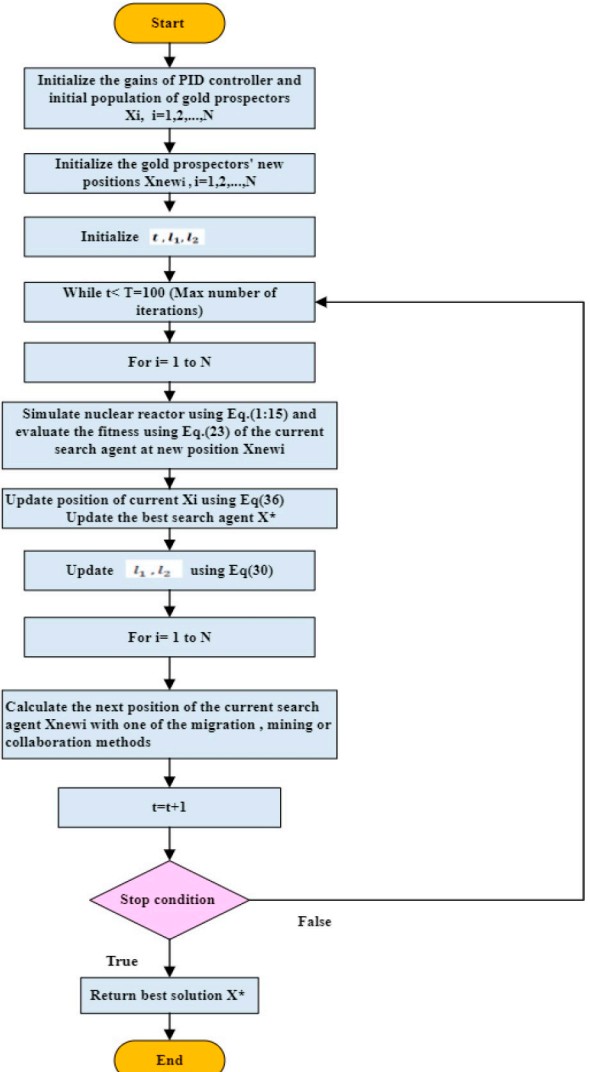

**Fig 3. Flowchart of the GRO algorithm.**

made as follows:

$$0 < k_P \leq 2,\ 0 < k_I \leq 0.1,\ \ 0 < k_D \leq 0.05$$

To enhance precision and attain optimal outcomes, the DA, AOA, POA, and GRO algorithms were executed 100 times. The optimum value of the OF is characterized as the minimum value obtained from the 100 iterations. Table 2 presents the PID gains' optimal values derived from nuclear model algorithms. Fig 4 depicts the convergence profile of the DA, AOA, POA, and GRO algorithms for the objective over 100 iterations. It is clear that the GRO meets the cost function with a lower iteration number and the cost value is nearly 0.5842 compared with the other comparable techniques, Furthermore, the GRO takes less time in minutes than other DA, POA, POA, and GRO algorithms for each iteration according to the Simulink model of the nuclear system.

**Table 1. Reactor design parameters.**

| Parameters | | Values | Parameters | | Values |
|---|---|---|---|---|---|
| Core diameter (inches) | | 119.7 | Coefficient of reactivity (1 /oF) | Moderator | -2.0 x 10$^{-4}$ |
| Core height (inches) | | 144 | | Fuel | - 1. 1 x 10$^{-5}$ |
| Delay neutron group fraction | First | 0.000209 | Prompt neutron generation time (sec) | | 1 .79 x x 10$^{-5}$ |
| | Second | 0.00141 4 | Nominal power output (mwt) | | 3436 |
| | Third | 0.001309 | Fraction of total power generated in fuel | | 0.974 |
| | Fourth | 0.002727 | Coolant volume ($ft^3$) | Upper plenum | 1376 |
| | Fifth | 0.000925 | | Lower plenum | 1 791 |
| | Sixth | 0.006898 | | Hot Leg pipings | 1 000 |
| | Total | 0.006898 | | Cold Leg pipings | 2000 |
| Group decay coefficient (1 /sec) | First | 0.0125 | | Core | 540 |
| | Second | 0.0308 | Mass of fuel (lbm) | | 222739 |
| | Third | 0.1 140 | Total coolant mass flow rate (lbm/hr) | | 1 .5 x 1 08 |
| | Fourth | 0.307 | Effective heat transfer area (Jt2) | | 59900 |
| | Fifth | 1. 19 | Specific heat capacity of fuel (btu\1bm—oF) | | 0.059 |
| | Sixth | 3.1 9 | Specific heat capacity of moderator (btu/1bm—oF) | | 1 .39 |
| | | | Average overall heat transfer coefficient (btuflbm—$ft^2$) | | 200 |

## Test Case 1: Constant power

In this scenario (as seen in Fig 5), the reference power is held steady at 1 pu, and the system is subjected to the suggested GRO-tuning PID controller as well as the other three optimization approaches utilized for comparison. When the proposed GRO methodology is used instead of the other three ways, the profile of the nuclear reactor core improves. When compared to the other 3 ways, the GRO-adjusting PID control strategy successfully minimized overshoots and attained the steady-state values of 1 pu quicker. Finally, it can be said that Fig 5 proves the efficacy of the proposed optimization method.

Fig 6 depicts the average, fuel, hotleg, and moderate temperatures in this regular operation utilizing the four optimized controllers. The suggested GRO-tuning PID controller provides the optimal response. The average temperature touched the maximum value of about 567.5 F° by GRO while it reached 568.5 F°, 567 F°, and 568 F° when applying POA, DA, and AOA techniques as seen in Fig 6(A), respectively.

## Test Case 2: Variable power

Another test case is run on the system to evaluate the effectiveness of the presented controller for regulating reactor power. In this situation, the reference power is suddenly reduced from 1 pu to 0.2 pu between 60 s and 140 s. The suggested GRO-tuning PID control strategy is employed and judged to three different optimization strategies for power regulation. When compared to the other three ways, the suggested GRO method performs the best. As shown in Fig 7, the response of the system's power owing to GRO has practically no overshoot and requires less settling time than DA, AOA, and POA-tuning approaches.

**Table 2. Tuning parameters of PID based on DA, AOA, POA, and GRO.**

| Parameter gain | DA | AOA | PA | GRO (proposed) |
|---|---|---|---|---|
| $k_P$ | 0.1201 | 0.0089 | 0.0377 | 0.5102 |
| $k_I$ | 2.0314 | 1.5024 | 2.1106 | 1.7300 |
| $k_D$ | 0.0000 | 0.0032 | 0.0276 | 0.0003 |

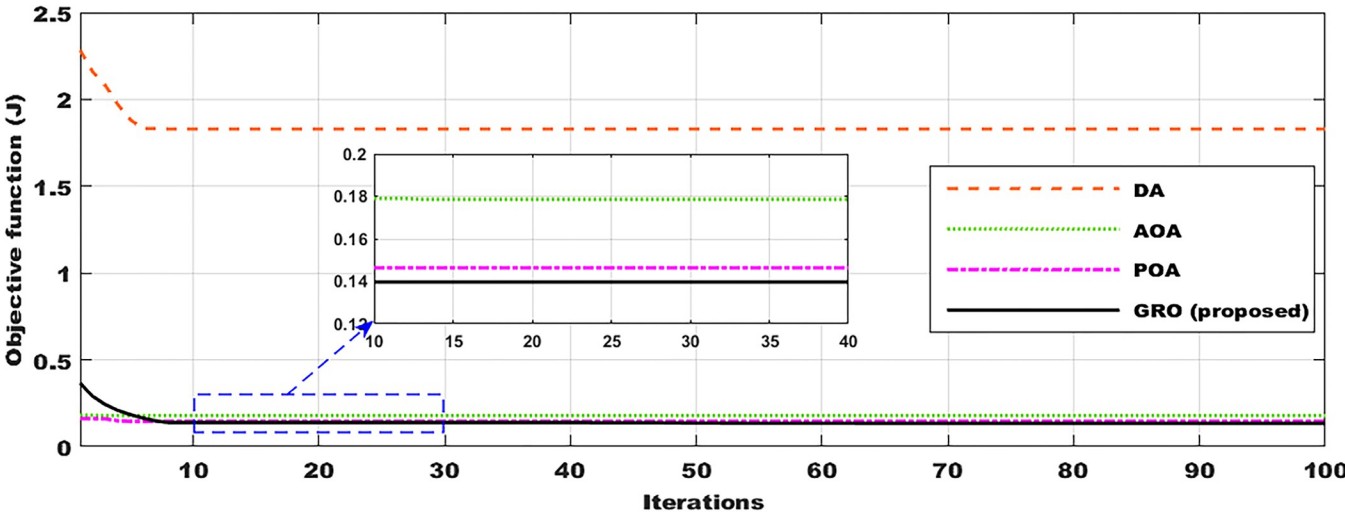

**Fig 4. Convergence curves with DA, AOA, POA, and GRO algorithms.**

With this quick shift in power, the average, hot leg, fuel, and moderate temperatures will drop, as shown in Fig 8(A)–8(D). The presented GRO-PID controller outperformed the other three optimization techniques. As shown in Fig 8, the temperature profile when using the suggested controller has practically no maximum overshoot and requires less setup time.

The findings mentioned in the second scenario demonstrate that when the power is adjusted, the suggested GRO-tuning PID controller provides the best performance. Table 3 provides a quantitative evaluation of system performance when the four algorithms are used. The GRO algorithm was shown to be highly successful when compared to DA, AOA, and POA-tuning PID controllers.

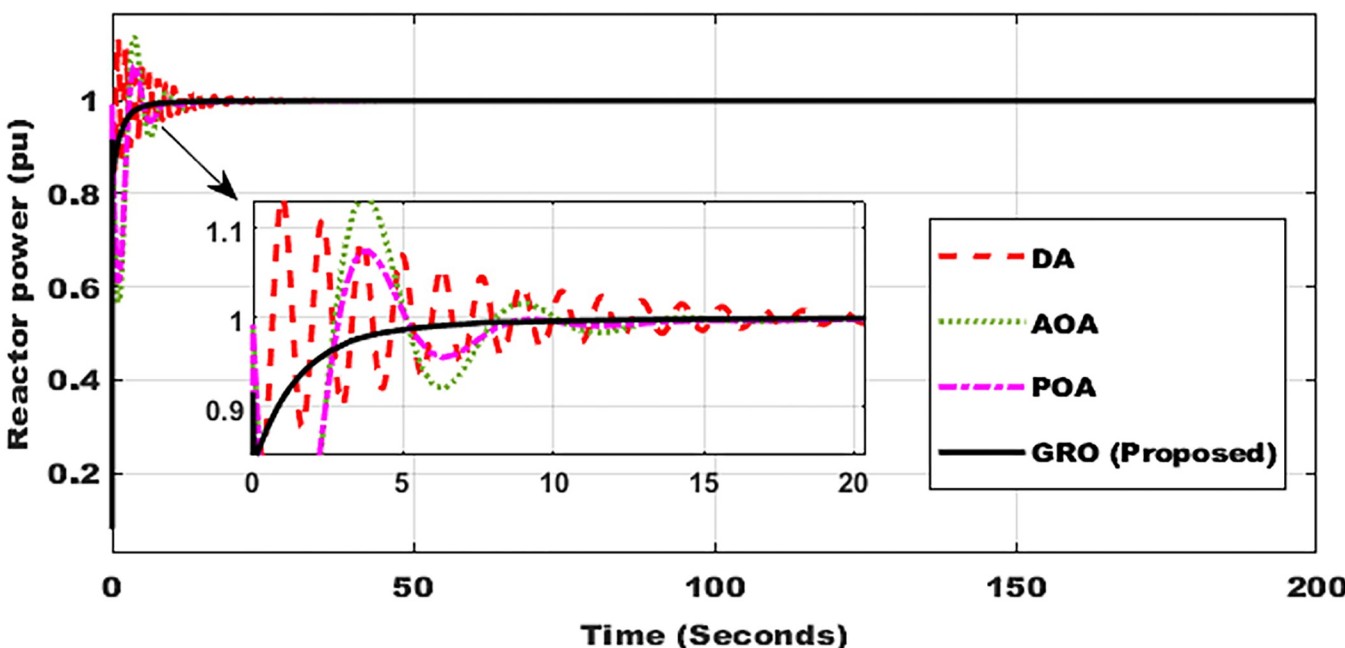

**Fig 5. Power response of nuclear reactor using DA, AOA, POA, and GRO algorithms.**

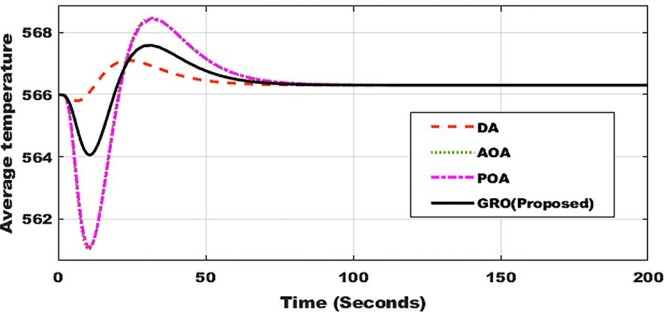

(a) Average temperature response.

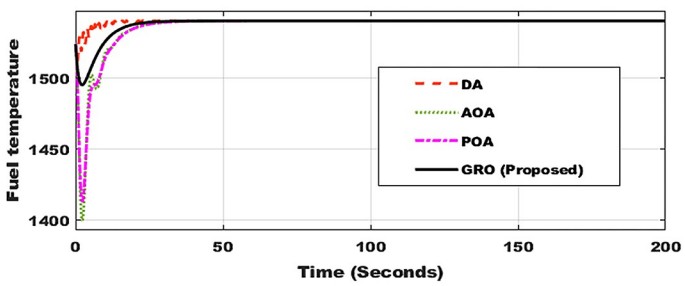

(b) Fuel temperature response.

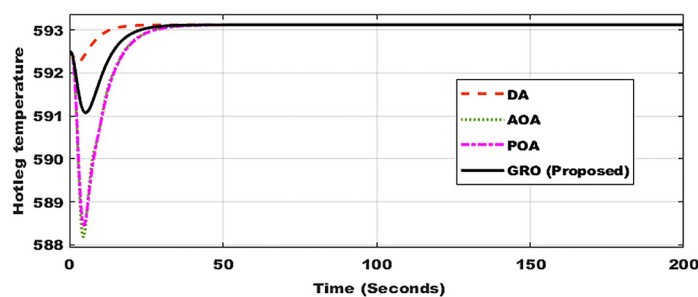

(c) Hotleg temperature response.

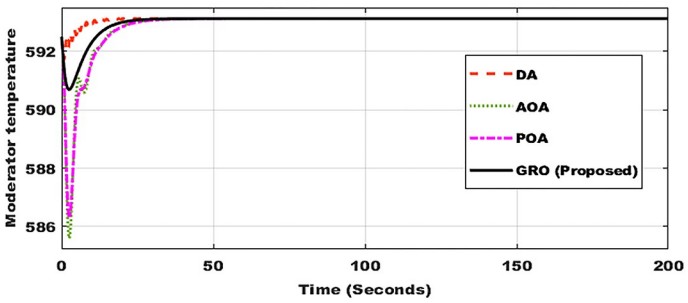

(d) Moderator temperature response.

**Fig 6. Temperatures profile utilizing DA, AOA, POA, and GRO algorithms.** (a) Average temperature response. (b) Fuel temperature response. (c) Hotleg temperature response. (d) Moderator temperature response.

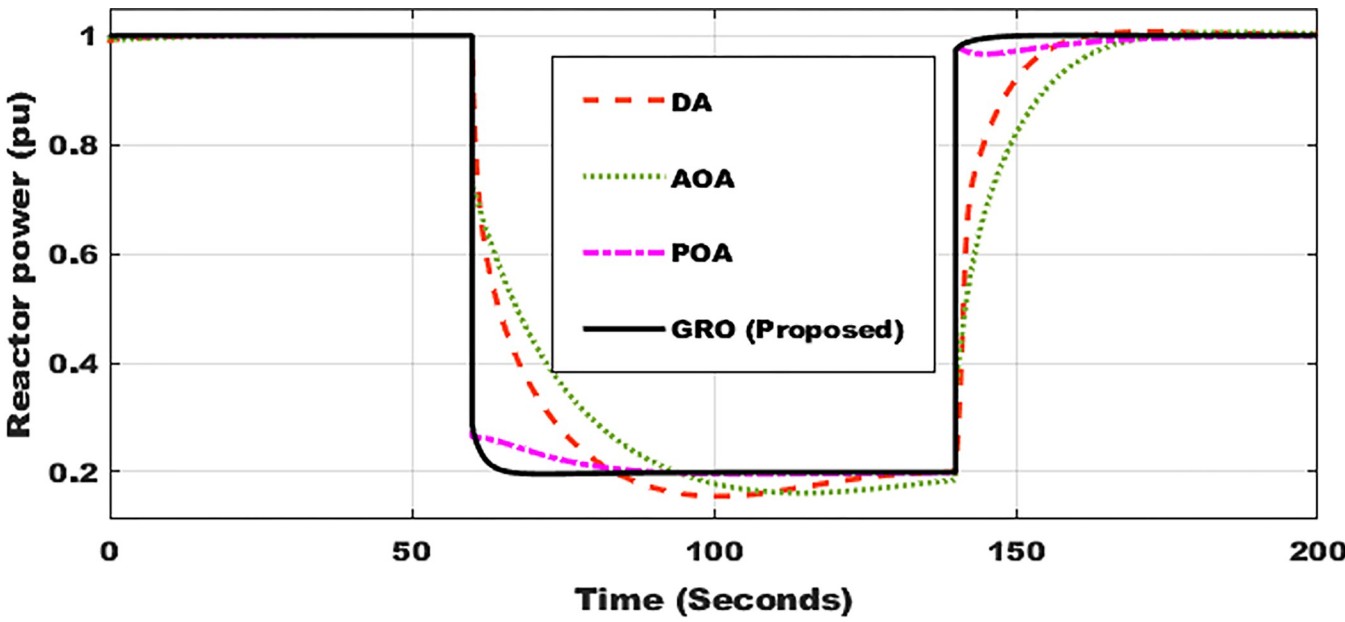

**Fig 7. Reactor power profile using DA, AOA, POA, and GRO algorithms.**

## 5. Conclusions

This paper conducts a thorough analysis of the steady-state and dynamic properties of PWRNPS in order to offer recommendations for the control system design and accomplish a broad range of flexible operations. This study introduces a new application of GRO, a meta-heuristic method used for adjusting and updating the parameters of PID controller gains. Based on a detailed analysis of PWRNPS's process characteristics, an optimized PID controller is constructed and confirmed to compensate for the absence of a dynamic model appropriate for control design. The controller's framework can be established by system analysis, and model settings can be obtained against the process data over a wide operating range. The non-linear controller that was created satisfies the specifications. This method is described for controlling a nuclear core reactor of the type PWR. This optimization strategy increased the performance of the nuclear system in terms of several control indices. The presented GRO--PID controller, as well as three additional algorithms (DA, AOA, and POA), are given and compared. The findings show that the suggested GRO algorithm outperformed the other three optimization approaches in terms of improving dynamic performance. Future studies will focus on improving the suggested controller's ability to disregard disturbances, as PWRNPS is invariably impacted by a variety of disturbances during practical operations. In addition, a few optimization strategies will be investigated to achieve optimal controller parameter tuning.

Future research directions:

1. Application of recent intelligent and robust controllers for the investigated system.

2. Investigation of the studied optimizers in other applications.

3. Integration of the investigated NE system with modern power systems, and studying its impacts and vice versa.

4. This system with the proposed optimizer may result in enhancing the power system stability under uncertainties, but this point needs more investigations.

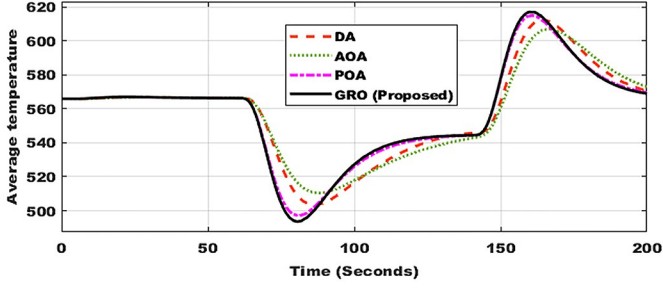

(a)   Average temperature response.

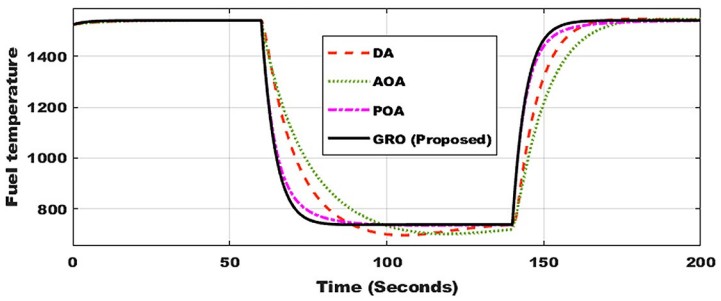

(b)   Fuel temperature.

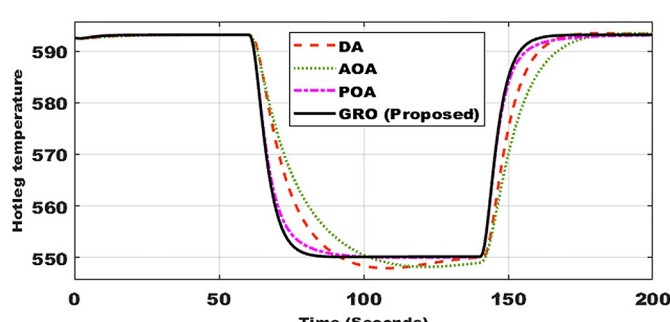

(c)   Hotleg temperature response.

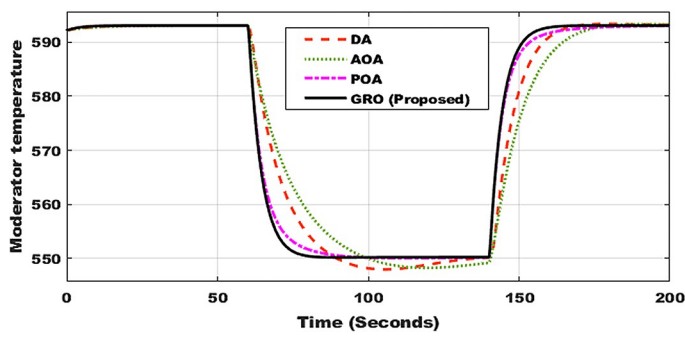

(d) Moderate temperature.

**Fig 8. Temperatures profile utilizing DA, AOA, POA, and GRO algorithms.** (a) Average temperature response. (b) Fuel temperature. (c) Hotleg temperature response. (d) Moderate temperature.

**Table 3. Relative study for 4 algorithms.**

| State variable | DA | | AOA | | POA | | GRO | |
|---|---|---|---|---|---|---|---|---|
| | % Max overshoot | Settling time (s) | % Max overshoot | Settling time (s) | % Max overshoot | Settling time (s) | % Max overshoot | Settling time (s) |
| Power | 0.1352 | 36.02 | 0.1358 | 26.66 | 0.0675 | 23.34 | 0.0095 | 17.95 |
| Fuel temperature | 0.0206 | 34.21 | 1.4034 | 47.10 | 1.2632 | 27.19 | 0.4400 | 23.22 |
| Hot leg temperature | 0.0015 | 44.31 | 0.0527 | 57.67 | 0.0452 | 41.84 | 0.0192 | 31.49 |
| Moderator temperature | 0.0014 | 42.86 | 0.0736 | 68.82 | 0.0661 | 34.15 | 0.0237 | 25.14 |

# Author Contributions

**Conceptualization:** Said A. kotb, Daniel Eutyche Mbadjoun Wapet.

**Investigation:** Said A. kotb, Daniel Eutyche Mbadjoun Wapet.

**Methodology:** Daniel Eutyche Mbadjoun Wapet.

**Resources:** Hany S. Hussein, Ahmed M. Ewais.

**Software:** H Abdelfattah, Daniel Eutyche Mbadjoun Wapet.

**Supervision:** Mohamed Metwally Mahmoud, Hany S. Hussein.

**Validation:** Daniel Eutyche Mbadjoun Wapet.

**Writing – original draft:** H Abdelfattah, Daniel Eutyche Mbadjoun Wapet.

**Writing – review & editing:** M Esmail, Mohamed Metwally Mahmoud, Daniel Eutyche Mbadjoun Wapet, Ahmed I. Omar, Ahmed M. Ewais.

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
