## [Decision Letter · Decision Letter 0]

9 Oct 2023

PONE-D-23-30160Optimal Controller Design for Reactor Core Power Stabilization in a Pressurized Water Reactor: Applications of Gold Rush AlgorithmPLOS ONE

Dear Dr. MBADJOUN WAPET,

Thank you for submitting your manuscript to PLOS ONE. After careful consideration, we feel that it has merit but does not fully meet PLOS ONE’s publication criteria as it currently stands. Therefore, we invite you to submit a revised version of the manuscript that addresses the points raised during the review process.

The authors must address all the comments of the reviewers and necessary changes must be made in the revised manuscript. 

We look forward to receiving your revised manuscript.

Kind regards,

Lalit Chandra Saikia, PhD

Academic Editor

PLOS ONE

Journal Requirements:

   "The authors extend their appreciation to the Deanship of Scientific Research at King Khalid University for funding this work through Small Groups Project under grant number (RGP.1/86 /44)."

6. Please ensure that you refer to Figure 5 in your text as, if accepted, production will need this reference to link the reader to the figure.

Additional Editor Comments:

The authors must address all the issues raised by the reviewers and necessary correction must be made in the revised manuscript.

Reviewers' comments:

Reviewer's Responses to Questions

**Comments to the Author**

1. Is the manuscript technically sound, and do the data support the conclusions?

Reviewer #1: Yes

Reviewer #2: Partly

2. Has the statistical analysis been performed appropriately and rigorously? 

Reviewer #1: Yes

Reviewer #2: N/A

3. Have the authors made all data underlying the findings in their manuscript fully available?

Reviewer #1: Yes

Reviewer #2: Yes

4. Is the manuscript presented in an intelligible fashion and written in standard English?

Reviewer #1: Yes

Reviewer #2: Yes

5. Review Comments to the Author

Reviewer #1: This manuscript discusses the evaluation and optimization of the pressurized water reactor (PWR) which is a dynamic model controlled by a PID controller optimized by the gold rush optimizer (GRO). The study concludes that the suggested optimized PID control strategy has superior efficiency and resilience in terms of less overshoot and settling time.

In general, the paper is well written. The state of the art is fine, and the included comparisons defending the goodness of the proposal against other existing and recent optimization methods are really good. This paper follows a clear and logical sequence. In addition, the contribution is complete and clear. However, I have a few suggestions to strengthen the paper further:

1. I suggest proofreading the manuscript, as there are several phrases that need improvements, for example:

The spelling of optimisation is a non-American variant. For consistency, consider replacing it with the American English spelling (optimization).

2. In section 4, the results and discussion must be accompanied by accurate and complete explanation and justification, for example:

Table 1, and figure 7 should be explained and discussed in more detail.

3. I suggest the authors identify the limitations of their approach.

4. . I suggest including a future work section, which could provide valuable insights and offer directions for further research.

Reviewer #2: This article is another example of the application of the GRO technique. There are some issues as follows:

1. Why do the authors need to use GRA for tuning the PID controller of the Reactor core power stabilization in a Pressurized Water Reactor? What are the motivations? In the introduction, the authors mention the reference [30] for the inspiration. The authors should explain more about this.

2. The authors claim that the GRA has many benefits over the other methods used in the comparison. Why did the authors select DA, AOA, and PA as comparison methods?

3. Why does the system's response in each row shown in Figure 8 look the same?

6. PLOS authors have the option to publish the peer review history of their article (what does this mean?). If published, this will include your full peer review and any attached files.

Reviewer #1: **Yes: **Mai Shawkat

Reviewer #2: No

---

## [Author Response · Author response to Decision Letter 0]

25 Oct 2023

***Technical response to the reviewers*** October 21th, 2023

Journal name: PLOS ONE

Manuscript No.: PONE-D-23-30160

Title: “Optimal Controller Design for Reactor Core Power Stabilization in a Pressurized Water Reactor: Applications of Gold Rush Algorithm”

H. Abdelfattah1, M. Esmail1, Said A. kotb2, Mohamed Metwally Mahmoud3, Hany S. Hussein4,5, Daniel Eutyche Mbadjoun Wapet6, *, Ahmed I. Omar7, Ahmed M. Ewais3

1 Electrical Department, Faculty of Technology and Education, Suez University, 41522, Suez, Egypt

2 Reactors Department, Egyptian Atomic Energy Authority (EAEA)

3Electrical Engineering Department, Faculty of Energy Engineering, Aswan University, Aswan 81528, Egypt

4Electrical Engineering Department, College of Engineering, King Khalid University, Abha 62529, Saudi Arabia

5Electrical Engineering Department, Aswan Faculty of Engineering, Aswan University, Aswan 81452, Egypt

6*National Advanced School of Engineering, Universit´e de Yaound´e I, Yaound´e, Cameroon

7 Electrical Power and Machines Engineering Department, The Higher Institute of Engineering at El-Shorouk City, El-Shorouk Academy, Cairo 11837, Egypt

hany.abdelfattah@ind.suezuni.edu.eg, m.e.hasanin55@gmail.com, saidabdou74@yahoo.com,

Metwally_M@aswu.edu.eg, hany.hussein@aswu.edu.eg, eutychedan@gmail.com, a.omar@sha.edu.eg

ewaisa@aswu.edu.eg

*Corresponding author: Daniel Eutyche Mbadjoun Wapet

Dear Editors and Reviewers

The authors are thankful to the learned Editor and Reviewers for their thoughtful and detailed comments to improve the quality of the manuscript. The authors have given reviewer comments a lot of interest in the revision process in an attempt to address all of the reviewers’ concerns and corrections as you will already find them incorporated in the revised manuscript. Moreover, a reply to each of the reviewers’ comments is provided below.

Kindly find the response to the reviewer’s comments in the following paragraphs. We hope this revised version of the manuscript meets the editor and reviewers’ expectations, and the standards of publication in the PLOS ONE Journal. 

The changes carried out by the authors are incorporated in the revised manuscript and highlighted in YELLOW.

Editor's Comments:

Comments to the Authors:

Comment-1: Thank you for submitting your manuscript to PLOS ONE. After careful consideration, we feel that it has merit but does not fully meet PLOS ONE’s publication criteria as it currently stands. Therefore, we invite you to submit a revised version of the manuscript that addresses the points raised during the review process. The authors must address all the comments of the reviewers and necessary changes must be made in the revised manuscript.

Response-1: Our sincere thanks and appreciation to the editor for considering our manuscript for publication in PLOS ONE Journal, and the recommending submission of the revised manuscript. To improve the quality of the manuscript, the reviewer's queries are addressed and their suggestions are incorporated into the revised manuscript. A new method called hybrid bat algorithm-balloon effect identifier optimizer is designed and implemented and compared with integral controller and Jaya+BE for frequency stability. The introduction section is rewritten, many changes in title, abstract, simulation results, and conclusions are done based on reviewers’ quires. Keywords are rearranged in aphetic order. Some sentences have been edited in the revised paper to clarify the paper's contributions and enhance the paper quality.

Comment-2: Please include the following items when submitting your revised manuscript:

Response-2: Our sincere thanks and appreciation to the editor for his comment. The required items are attached during submission process. A cover letter is provided and prepared to explain, point by point, the details of the revisions to the manuscript. The changes carried out by the authors are incorporated in the revised manuscript and highlighted in YELLOW to be easily viewed by the editors and reviewers. An unmarked version of the revised paper without tracked changes is also provided.

Comment-3: Please ensure that your manuscript meets PLOS ONE's style requirements.

Response-3: The authors are extremely thankful to the editor for this thoughtful point. The revised manuscript meets PLOS ONE's style.

Reviewers Comments:

Reviewer 1 

Comments to the Authors:

Comment-1: This manuscript discusses the evaluation and optimization of the pressurized water reactor (PWR) which is a dynamic model controlled by a PID controller optimized by the gold rush optimizer (GRO). The study concludes that the suggested optimized PID control strategy has superior efficiency and resilience in terms of less overshoot and settling time.

In general, the paper is well written. The state of the art is fine, and the included comparisons defending the goodness of the proposal against other existing and recent optimization methods are really good. This paper follows a clear and logical sequence. In addition, the contribution is complete and clear. However, I have a few suggestions to strengthen the paper further:

1. I suggest proofreading the manuscript, as there are several phrases that need improvements, for example: The spelling of optimisation is a non-American variant. For consistency, consider replacing it with the American English spelling (optimization).

Response-1: At the beginning, the authors are thankful to the honorable reviewer for the words of encouragement and trust in our work. As suggested by the esteemed reviewer, proofreading is done in the updated manuscript. The authors apologize regarding these typos. Kindly check the revised manuscript.

Comment-2: In section 4, the results and discussion must be accompanied by accurate and complete explanation and justification, for example: Table 1, and figure 7 should be explained and discussed in more detail.

Response-2: The authors are thankful to the esteemed reviewer upon his valuable comment. Based on the reviewer comment, Table 1, and Figure 7 have been explained and discussed in more detail in the updated paper. All the reviewer suggestions are done in the updated paper. Kindly check the revised manuscript.

Comment-3: I suggest the authors identify the limitations of their approach.

Response-3: The authors are thankful to the esteemed reviewer upon his valuable comment. The needs and benefits of this study is provided in the updated paper. The proposed method did not appear in any published research article, so it is needed to investigate this method performance.

Despite the good response achieved by using algorithms, the door is opened to more enhancements in system responses during difficulties such as parameters changes. Addition explanation has been added to clarify the need of the proposed algorithm. Kindly check the revised manuscript.

Comment-4: I suggest including a future work section, which could provide valuable insights and offer directions for further research.

Response-4: The authors are thankful to the esteemed reviewer upon his valuable comment. Based on the esteemed reviewer suggestion, including a future work section is done in the updated paper. Kindly check the revised manuscript.

Mai Shawkat add at least 3 references.

Reviewer 2 

Comment-1: This article is another example of the application of the GRO technique. There are some issues as follows: 1. Why do the authors need to use GRA for tuning the PID controller of the Reactor core power stabilization in a Pressurized Water Reactor? What are the motivations? In the introduction, the authors mention the reference [30] for the inspiration. The authors should explain more about this.

Response-1: At the beginning, the authors are thankful to the honorable reviewer for the words of encouragement and trust in our work. The authors need to use GRA for tuning the PID controller of the reactor core power stabilization in a pressurized water reactor to test this algorithm effectiveness in enhancing the system dynamic performance. The motivations of this study are highlighted in the updated paper. More explanation is provided for the mentioned reference in the updated paper. Kindly check the revised manuscript.

Comment-2: The authors claim that the GRA has many benefits over the other methods used in the comparison. Why did the authors select DA, AOA, and PA as comparison methods?

Response-2: The authors are thankful to the esteemed reviewer upon his valuable comment. The authors are thankful to the esteemed reviewer upon his valuable comment. Many industrial applications have employed the DA, AOA, and PA to adaptively adjust the gains of traditional controllers, so these are considered for comparison. 

Comment-3: Why does the system's response in each row shown in Figure 8 look the same?

Response-3: The authors are extremely thankful to the reviewer for this thoughtful point. The system's response in each row shown in Figure 8 look the same is the normal because the changed is the controller setting only. However, in this figure the proposed algorithm proves its accuracy and effectiveness. Kindly check the revised manuscript. 

The authors once again thank the learned Editors and Reviewers for their valuable comments for improving the quality of the manuscript.

---

## [Decision Letter · Decision Letter 1]

16 Nov 2023

PONE-D-23-30160R1Optimal Controller Design for Reactor Core Power Stabilization in a Pressurized Water Reactor: Applications of Gold Rush AlgorithmPLOS ONE

Dear Dr. Mbadjoun Wapet,

Thank you for submitting your manuscript to PLOS ONE. After careful consideration, we feel that it has merit but does not fully meet PLOS ONE’s publication criteria as it currently stands. Therefore, we invite you to submit a revised version of the manuscript that addresses the points raised during the review process.

The reviewers forwarded the comments. The authors must addressed all the comments of the reviewers and modify the manuscript.

We look forward to receiving your revised manuscript.

Kind regards,

Lalit Chandra Saikia, PhD

Academic Editor

PLOS ONE

Reviewers' comments:

Reviewer's Responses to Questions

**Comments to the Author**

1. If the authors have adequately addressed your comments raised in a previous round of review and you feel that this manuscript is now acceptable for publication, you may indicate that here to bypass the “Comments to the Author” section, enter your conflict of interest statement in the “Confidential to Editor” section, and submit your "Accept" recommendation.

Reviewer #2: All comments have been addressed

Reviewer #3: (No Response)

2. Is the manuscript technically sound, and do the data support the conclusions?

Reviewer #2: Yes

Reviewer #3: (No Response)

3. Has the statistical analysis been performed appropriately and rigorously? 

Reviewer #2: Yes

Reviewer #3: (No Response)

4. Have the authors made all data underlying the findings in their manuscript fully available?

Reviewer #2: Yes

Reviewer #3: (No Response)

5. Is the manuscript presented in an intelligible fashion and written in standard English?

Reviewer #2: Yes

Reviewer #3: (No Response)

6. Review Comments to the Author

Reviewer #2: I have no more issues with the paper. The paper is well written. The authors have improved the paper according to the reviwer's comment.

Reviewer #3: Appreciating and publishable work but requires few improvements. The paper should be further revised by incorporating the following comments:

1. Refine abstract. Abstract should be improved first explaining a problem and then explaining its proposed solution and benefits over existing strategies.

2. Introduction should also include Research gap and motivation, Challenges, and Contribution (in points).

3. The main contributions of this paper should be further summarized and clearly demonstrated. This reviewer suggests the authors exactly mention what is new compared with existing approaches and why the proposed approach is needed to be used instead of the existing methods.

4. PID is not a new controller.

5. Objective function (OF) is not new.

6. All result lines should not be solid. Differentiate them via solid, dashed, dot, dot-dashed etc.

7. More results should be given.

8. Results with different nonlinearities should be given.

9. Add corresponding reference to tables/figures.

10. Robustness of the method should also be validated in frequency/time domain.

11. Compare the results with existing/published results.

12. Literature review should be more strengthen by adding few more papers like

https://doi.org/10.1002/2050-7038.12883;
https://doi.org/10.1016/j.jestch.2020.12.023;

https://doi.org/10.17341/gazimmfd.841751.

7. PLOS authors have the option to publish the peer review history of their article (what does this mean?). If published, this will include your full peer review and any attached files.

Reviewer #2: No

Reviewer #3: No

---

## [Author Response · Author response to Decision Letter 1]

24 Nov 2023

***Technical response to the reviewers*** November 23th, 2023

Journal name: PLOS ONE

Manuscript No.: PONE-D-23-30160

Title: “Optimal Controller Design for Reactor Core Power Stabilization in a Pressurized Water Reactor: Applications of Gold Rush Algorithm”

H. Abdelfattah1, M. Esmail1, Said A. kotb2, Mohamed Metwally Mahmoud3, Hany S. Hussein4,5, Daniel Eutyche Mbadjoun Wapet6, *, Ahmed I. Omar7, Ahmed M. Ewais3

1 Electrical Department, Faculty of Technology and Education, Suez University, 41522, Suez, Egypt

2 Reactors Department, Egyptian Atomic Energy Authority (EAEA)

3Electrical Engineering Department, Faculty of Energy Engineering, Aswan University, Aswan 81528, Egypt

4Electrical Engineering Department, College of Engineering, King Khalid University, Abha 62529, Saudi Arabia

5Electrical Engineering Department, Aswan Faculty of Engineering, Aswan University, Aswan 81452, Egypt

6*National Advanced School of Engineering, Universit´e de Yaound´e I, Yaound´e, Cameroon

7 Electrical Power and Machines Engineering Department, The Higher Institute of Engineering at El-Shorouk City, El-Shorouk Academy, Cairo 11837, Egypt

hany.abdelfattah@ind.suezuni.edu.eg, m.e.hasanin55@gmail.com, saidabdou74@yahoo.com,

Metwally_M@aswu.edu.eg, hany.hussein@aswu.edu.eg, eutychedan@gmail.com, a.omar@sha.edu.eg

ewaisa@aswu.edu.eg

*Corresponding author: Daniel Eutyche Mbadjoun Wapet

Dear Editors and Reviewers

The authors are thankful to the learned Editor and Reviewers for their thoughtful and detailed comments to improve the quality of the manuscript. The authors have given reviewer comments a lot of interest in the revision process in an attempt to address all of the reviewers’ concerns and corrections as you will already find them incorporated in the revised manuscript. Moreover, a reply to each of the reviewers’ comments is provided below.

Kindly find the response to the reviewer’s comments in the following paragraphs. We hope this revised version of the manuscript meets the editor and reviewers’ expectations, and the standards of publication in the PLOS ONE Journal. 

The changes carried out by the authors are incorporated in the revised manuscript and highlighted in YELLOW.

Editor's Comments:

Comments to the Authors:

Comment-1: Thank you for submitting your manuscript to PLOS ONE. After careful consideration, we feel that it has merit but does not fully meet PLOS ONE’s publication criteria as it currently stands. Therefore, we invite you to submit a revised version of the manuscript that addresses the points raised during the review process. The authors must address all the comments of the reviewers and necessary changes must be made in the revised manuscript. The reviewers forwarded the comments. The authors must address all the comments of the reviewers and modify the manuscript.

Response-1: Our sincere thanks and appreciation to the editor for considering our manuscript for publication in PLOS ONE Journal, and the recommending submission of the revised manuscript. To improve the quality of the manuscript, the reviewer's queries are addressed and their suggestions are incorporated into the revised manuscript. Some sentences have been edited in the revised paper to clarify the paper's contributions and enhance the paper quality.

Comment-2: Please include the following items when submitting your revised manuscript:

Response-2: Our sincere thanks and appreciation to the editor for his comment. The required items are attached during submission process. A cover letter is provided and prepared to explain, point by point, the details of the revisions to the manuscript. The changes carried out by the authors are incorporated in the revised manuscript and highlighted in YELLOW to be easily viewed by the editors and reviewers. An unmarked version of the revised paper without tracked changes is also provided.

Comment-3: Please ensure that your manuscript meets PLOS ONE's style requirements.

Response-3: The authors are extremely thankful to the editor for this thoughtful point. The revised manuscript meets PLOS ONE's style.

Reviewers Comments:

Reviewer 2 

Comments to the Authors:

Comment-2: I have no more issues with the paper. The paper is well written. The authors have improved the paper according to the reviewer’s comment.

Response-1: At the beginning, the authors are thankful to the honorable reviewer for the words of encouragement and trust in our work. 

Reviewer 3 

Comment-1: Appreciating and publishable work but requires few improvements. The paper should be further revised by incorporating the following comments:

1. Refine abstract. The abstract should be improved first by explaining a problem and then explaining its proposed solution and benefits over existing strategies.

Response-1: At the beginning, the authors are thankful to the honorable reviewer for the words of encouragement and trust in our work. Based on the esteemed reviewer's suggestion, the abstract is improved in the updated paper. Kindly check the revised manuscript.

Comment-2: The introduction should also include the Research gap and motivation, Challenges, and Contribution (in points).

Response-2: The authors are thankful to the esteemed reviewer for his valuable comment. The introduction part of the updated paper includes the suggested items. Kindly check the revised manuscript.

Comment-3: The main contributions of this paper should be further summarized and clearly demonstrated. This reviewer suggests the authors exactly mention what is new compared with existing approaches and why the proposed approach is needed to be used instead of the existing methods.

Response-3: The authors are extremely thankful to the reviewer for this thoughtful point. The authors need to use GRA for tuning the PID controller of the reactor core power stabilization in a pressurized water reactor to test this algorithm effectiveness in enhancing the system dynamic performance. The contributions of this study are highlighted in the updated paper. Kindly check the revised manuscript.

Comment-4: PID is not a new controller.

Response-4: The authors are extremely thankful to the reviewer for this thoughtful point. The PID controller is used in industrial till now, and we apply a new optimization. Kindly check the revised manuscript. 

Comment-5: Objective function (OF) is not new.

Response-5: The authors are extremely thankful to the reviewer for this thoughtful point. The three parameters of the PID controller (K_P,K_I,K_D) are optimized in this work using GRO to boost the NE reactor's performance during load-following operation. In order to gauge how well the suggested optimization method performs, a comparison was provided between DA, AOA, PA, and GRO algorithms.

As shown in Eq. (23), the objective function (OF) is a set of time-domain integral performance indices. The main purpose is to estimate the minimum values of an index function in order to determine the best PID control strategy settings. The integral of time-weighted absolute error (ITAE) motivates the selection of an objective function that seeks to minimize the temporal response characteristics. Furthermore, this OF proves its accuracy and effectiveness in different engineering problems.

 J_1=ITAE=∫_0^Ts▒t|e(t)|dt (23)

Comment-6: All result lines should not be solid. Differentiate them via solid, dashed, dot, dot-dashed etc.

Response-6: The authors are extremely thankful to the reviewer for this thoughtful point. Based on the esteemed reviewer suggestion, differentiate all result lines via solid, dashed, dot, dot-dashed etc. is performed in the updated paper. Kindly check the revised manuscript.

Comment-7: More results should be given.

Response-7: The authors are extremely thankful to the reviewer for this thoughtful point. Two test scenarios are provided to evaluate the suggested controller's efficacy. The first scenario involves constant reference power, whereas the second involves fluctuating reference power. A comparison through four various optimization methods is introduced to judge the presented GRO-tuned PID controller. These four optimizations-PID controllers are DA, AOA, PA, and GRO. In addition, future research directions are provided at the end of this paper. Kindly check the revised manuscript. 

Comment-8: Results with different nonlinearities should be given.

Response-8: The authors are extremely thankful to the reviewer for this thoughtful point. The second scenario involves fluctuating reference power to show the superiority of the proposed optimized controller. Kindly check the revised manuscript. 

Comment-9: Add corresponding reference to tables/figures.

Response-9: The authors are extremely thankful to the reviewer for this thoughtful point. Adding the corresponding reference to tables/figures is done in the updated paper. Kindly check the revised manuscript. 

Comment-10: Robustness of the method should also be validated in frequency/time domain.

Response-10: The authors are extremely thankful to the reviewer for this thoughtful point. Two test scenarios are provided to evaluate the suggested controller's efficacy. The first scenario involves constant reference power, whereas the second involves fluctuating reference power. A comparison through four various optimization methods is introduced to judge the presented GRO-tuned PID controller. These four optimizations-PID controllers are DA, AOA, PA, and GRO. In this study, the reactor core of the PWR scheme is utilized simultaneously through the controllers shown utilizing MATLAB/SIMULINK.

Comment-11: Compare the results with existing/published results.

Response-11: The authors are extremely thankful to the reviewer for this thoughtful point. Comparing the results with published works is presented in the literature review part. Kindly check the revised manuscript. 

Comment-12: Literature review should be more strengthen by adding few more papers like

https://doi.org/10.1002/2050-7038.12883;
https://doi.org/10.1016/j.jestch.2020.12.023;

https://doi.org/10.17341/gazimmfd.841751.

Response-12: The authors are extremely thankful to the reviewer for this thoughtful point. The suggested references are helful and strength the literature review part. Kindly check the revised manuscript. 

 G. Sharma, N. Krishnan, Y. Arya, and A. Panwar, “Impact of ultracapacitor and redox flow battery with JAYA optimization for frequency stabilization in linked photovoltaic-thermal system,” Int. Trans. Electr. Energy Syst., vol. 31, no. 5, 2021, doi: 10.1002/2050-7038.12883.

 I. Eke, M. Saka, H. Gozde, Y. Arya, and M. C. Taplamacioglu, “Heuristic optimization based dynamic weighted state feedback approach for 2DOF PI-controller in automatic voltage regulator,” Eng. Sci. Technol. an Int. J., vol. 24, no. 4, pp. 899–910, 2021, doi: 10.1016/j.jestch.2020.12.023.

 Z. YENEN YILMAZ, G. BAL, E. ÇELİK, N. OZTURK, U. GÜVENÇ, and Y. ARYA, “Yük frekans kontrolünde kullanılan ikincil denetleyicilerin optimizasyonuna yönelik yeni bir hedef fonksiyonu tasarımı,” Gazi Üniversitesi Mühendislik Mimar. Fakültesi Derg., vol. 36, no. 4, pp. 2053–2068, 2021, doi: 10.17341/gazimmfd.841751.

The authors once again thank the learned Editors and Reviewers for their valuable comments for improving the quality of the manuscript.

---

## [Decision Letter · Decision Letter 2]

28 Dec 2023

Optimal Controller Design for Reactor Core Power Stabilization in a Pressurized Water Reactor: Applications of Gold Rush Algorithm

PONE-D-23-30160R2

Dear Dr. Mbadjoun Wapet,

We’re pleased to inform you that your manuscript has been judged scientifically suitable for publication and will be formally accepted for publication once it meets all outstanding technical requirements.

Kind regards,

Lalit Chandra Saikia, PhD

Academic Editor

PLOS ONE

Additional Editor Comments (optional):

The paper is provisionally accepted.

Reviewers' comments:

Reviewer's Responses to Questions

**Comments to the Author**

1. If the authors have adequately addressed your comments raised in a previous round of review and you feel that this manuscript is now acceptable for publication, you may indicate that here to bypass the “Comments to the Author” section, enter your conflict of interest statement in the “Confidential to Editor” section, and submit your "Accept" recommendation.

Reviewer #2: All comments have been addressed

Reviewer #3: (No Response)

2. Is the manuscript technically sound, and do the data support the conclusions?

Reviewer #2: Yes

Reviewer #3: (No Response)

3. Has the statistical analysis been performed appropriately and rigorously? 

Reviewer #2: Yes

Reviewer #3: (No Response)

4. Have the authors made all data underlying the findings in their manuscript fully available?

Reviewer #2: Yes

Reviewer #3: (No Response)

5. Is the manuscript presented in an intelligible fashion and written in standard English?

Reviewer #2: Yes

Reviewer #3: (No Response)

6. Review Comments to the Author

Reviewer #2: I have no more issues with the paper. The authors have corrected all the problems according to the reviewer's comments.

Reviewer #3: No more comments.

The authors have incorporated all the suggestions offered by this reviewer.

The paper is now in acceptable form.

7. PLOS authors have the option to publish the peer review history of their article (what does this mean?). If published, this will include your full peer review and any attached files.

Reviewer #2: No

Reviewer #3: No

---

## [Editor Report · Acceptance letter]

17 Jan 2024

PONE-D-23-30160R2 

PLOS ONE

Dear Dr. Mbadjoun Wapet, 

I'm pleased to inform you that your manuscript has been deemed suitable for publication in PLOS ONE. Congratulations! Your manuscript is now being handed over to our production team.

Kind regards, 

on behalf of

Dr. Lalit Chandra Saikia 

Academic Editor

PLOS ONE